# Role of the Intestinal Microbiota in the Molecular Pathogenesis of Atypical Parkinsonian Syndromes

**DOI:** 10.3390/ijms26093928

**Published:** 2025-04-22

**Authors:** Dominika Przewodowska, Piotr Alster, Natalia Madetko-Alster

**Affiliations:** Department of Neurology, Medical University of Warsaw, Kondratowicza 8, 03-242 Warsaw, Poland; piotr.alster@wum.edu.pl

**Keywords:** atypical Parkinsonian syndromes, Parkinson disease, neurodegeneration, intestinal microbiota, systemic inflammation, gut–brain axis, α-synuclein, tau, short-chain fatty acids, vitamin B12

## Abstract

The role of the intestinal microbiota and its influence on neurodegenerative disorders has recently been extensively explored, especially in the context of Parkinson’s disease (PD). In particular, its role in immunomodulation, impact on inflammation, and participation in the gut–brain axis are under ongoing investigations. Recent studies have revealed new data that could be important for exploring the neurodegeneration mechanisms connected with the gut microbiota, potentially leading to the development of new methods of treatment. In this review, the potential roles of the gut microbiota in future disease-modifying therapies were discussed and the properties of the intestinal microbiota—including its impacts on metabolism and short-chain fatty acids and vitamins—were summarized, with a particular focus on atypical Parkinsonian syndromes. This review focused on a detailed description of the numerous mechanisms through which the microbiota influences neurodegenerative processes. This review explored potentially important connections between the gut microbiota and the evolution and progression of atypical Parkinsonian syndromes. Finally, a description of recently derived results regarding the microbiota alterations in atypical Parkinsonian syndromes in comparison with results previously described in PD was also included.

## 1. Introduction

Atypical Parkinsonian syndromes (APSs) comprise various neurodegenerative disorders characterized by rapid progression, poor levodopa responsiveness and shorter life expectancy when compared to Parkinson’s disease (PD) [1]. Among them, Progressive Supranuclear Palsy (PSP), Multiple System Atrophy (MSA), Corticobasal Syndrome (CBS) and Frontotemporal Dementia (FTD) can be distinguished. Although all of the above entities are classified as neurodegenerative diseases caused by the accumulation of abnormal proteins, the compositions of these deposits differ. PSP and CBS are grouped with the FTD-related disorders due to their underlying tau pathology, while MSA and PD are described as α-synucleinopathies [2].

Frontotemporal dementia (FTD), first described by Pick in 1892, is a clinical manifestation of neuronal degeneration in the frontal and temporal lobes. As these structures are mainly responsible for speech and behavior control, loss of neurons leads to progressive aphasia and cognitive and behavioral dysfunction [3]. Pathologically, it is recognized by a tau- and TDP-43-based pathology [4]. In some cases, patients initially diagnosed with FTD develop additional symptoms, such as gait disturbance, increased falls and supranuclear palsy [4]. In this case, the behavioral FTD variant of PSP should be diagnosed (PSP-bvFTD). It is worth mentioning that, among the various PSP subtypes, PSP-Parkinsonism predominant (PSP-P) and PSP-Richardson Syndrome (PSP-RS) are the most common. Pseudobulbar and horizontal gaze palsy concomitant with axial rigidity and postural instability are described as the core clinical features of PSP. In PSP-P, tremor, rigidity and bradykinesia are typical symptoms, while a reduction in horizontal eye movement, gait instability, falls and profoundly progressing cognitive decline is described in the context of PSP-RS [4].

The main symptom of MSA is dysautonomia (dysfunction of the autonomic system) concomitant with cerebellar signs (MSA-C) or Parkinsonism (MSA-P) [5]. The chronic progression of clinical symptoms with transient, poor or no levodopa response and relatively rare cognitive decline are characteristic of the disease.

Neurocognitive dysfunction—often misdiagnosed as AD—is a common sign of CBS. Coincidentally, it is also characterized by alien limb phenomena, described as asymmetric rigidity, hypokinesia and dystonia with numerous uncontrolled limb movements [2]. At present, there are no data concerning dysbiosis in corticobasal degeneration (CBD), which could be (at least partially) explained by the rarity of the disease and the need for neuropathological examination in order to obtain a diagnosis. CBS, which can be diagnosed based on the clinical syndrome, is very diverse in the context of neuropathology, which could explain the lack of studies concerning the gut microbiota with respect to this disease.

The criteria for diagnosing particular forms of atypical Parkinsonism are constantly being modified in response to the results of new studies. The final and only certain way to make the right diagnosis is a post-mortem examination of the brain. Making the most probable diagnosis as soon as possible is crucial, as standard methods of therapy are not satisfactory in patients with APSs. Modern researchers are trying to find as many potential factors as possible that contribute to neurodegeneration in patients with APSs in the hope of determining possible drug targets for use in future therapies. The data from APS studies have indicated differences in the type of protein accumulated in the CNS, the levels of individual inflammatory markers, the degree of reaction to pain, and the expression levels of genes leading to the development of pathology in the brain [3,4,6,7].

While the prevalence of Parkinson’s disease is expected to increase from 76 (in 2021) to 267 cases per 100,000 in 2050, the prevalence of APS tends to remain stable over time and has been estimated as less than 10 per 10,000 [8,9]. At present, the exact pathomechanism of neurodegeneration in APS remains unknown, although there is likely a multitudinous combination of endo- and exogenic factors generally affecting the pace of neurodegeneration [10]. The genetic background of neuronal deterioration, as well as its correlation with toxin exposure or brain injury, have been broadly described in the literature, although an “enriched environment” due to anti-oxidative properties seems to act as another factor impacting the number of neuronal cell deaths [10,11]. Among multiple theories explaining the mechanisms of neurodegeneration, the involvement of inflammatory processes is certain, although it has still not been definitively established whether it this a cause or result of the disease. Recognizing the human body as a combination of interconnected systems, the search for factors influencing the exacerbation of inflammation is one of the most widespread topics in the contemporary literature.

Although Braak’s dual-hit hypothesis of sporadic PD development has been widely recognized for nearly 20 years, its accuracy remains a subject of debate. According to Braak’s idea, the process of neurodegeneration within the central nervous system (CNS) remains strictly dependent on the conditions of the digestive tract, being the expanded system connecting the external environment with the brain. When potential pathogens (virus, bacteria) achieve access to the olfactory bulb, they can also pass through the nasal cavity and into the digestive tract after being swallowed. Through the promotion of regional inflammation and intestinal barrier disruption, these pathogens stimulate local α-synuclein accumulation, which then could be transported by the vagal nerve into the CNS [12]. Braak’s theory remains only partially in line with that of Horsager, who has proposed another mechanism of PD development and highlighted its two phenotypes [13]. While Braak’s theory applies to PD and α-synuclein, it is not known whether and to what extent it can be extended to tauopathies. The previously mentioned model of α-synuclein accumulation starts with non-specific inflammation in the gastrointestinal tract and assumes its further dispersion into the CNS. According to Horsager, this is described as a body-first phenotype, linked with an increased risk of motor/dysautonomic symptoms, clinically distinguished from the second, brain-first phenotype by the presence of rapid eye movement (REM) sleep behavior disorder (RBD). According to this theory, the disease can start initially in the CNS—on the level of the substantia nigra pars compacta dopaminergic neurons—and then spread within the cortex. This model is more common in patients with *LRRK2* mutations [14]. Due to clinical similarities, it could be assumed that the mechanisms underlying APSs might resemble one another. Beside the fact that inflammation is present even before protein accumulation, the number of involved immunological pathways is so wide that it seems to be impossible to choose only one factor strictly related to the initiation of inflammatory processes and eventual progression. While diet, aging, infections and gut dysbiosis (imbalance in the gut microflora), among other factors, seem to represent environmental modulators, the roles of genetics and endogenic mediators have also been strictly discussed.

According to McGeers’ hypothesis, a local inflammatory reaction within the brain can promote microglial activation, resulting in enhanced oxidative stress reactions. Increased expression levels of IL-1, IL-6 and TNF-alpha in cerebral tissue concurrent with increased complement system activity provides evidence for the possible role of regional inflammation when explaining the pathogenesis of PD [15]. The exact role of inflammation in initiating neurodegeneration or its potential reactive character still requires more extensive evaluation.

The existence of neuro-inflammatory processes in neurodegenerative disorders is not in doubt and has been observed in in vivo trials [16]. Furthermore, multiple studies have provided data supporting the potential role of inflammation in the course of the disease; for example, Madetko-Alster et al. described differences between the cytokine profiles in blood and CSF among patients with PSP-P and PSP-RS. Significant differences between IL-1 and IL-6 concentrations with respect to PSP-P and PSP-RS patients could be explained by microglial activity and IL release, with a potential impact on the progression of the disease [17].

The inflammatory processes present during neurodegeneration are generalized and can be reflected by unspecific peripheral parameters. Investigations of the serum lymphocyte-to-monocyte ratio (LMR), neutrophil-to-lymphocyte ratio (NLR) and neutrophil-to-high density lipoprotein ratio (NHR) revealed higher parameters of NLR and NHR concurrently with lower LMR in patients with PD in comparison with HC, in correspondence with the severity of the disease [18]. Madetko-Alster et al. reported increased levels of NLR as well as another parameter—the platelet-to-lymphocyte ratio (PLR)—in PD (in comparison with HC), with no statistically significant differences in NLR and PLR between patients with PD and MSA-P. In MSA-P patients, only NLR was significantly higher when compared with HC [19]. Although no explanation for this observation has been fully established, the intestinal microbiota and its bacterial composition could act as potential factors triggering inflammation, thus resulting in different levels of inflammatory enhancement.

This paper provides a narrative review concerning the potential roles of the gut microbiota in the pathogenesis of atypical Parkinsonian syndromes. The PubMed and Google Scholar databases were searched for papers related to the molecular mechanisms of bacterial activity in the context of atypical Parkinsonian syndromes. The following medical subject headings were used: “atypical parkinsonian syndromes—role of intestinal microbiota”, “molecular mechanisms of neurodegeneration”, “microbiota activity”, “gut-brain axis activity”, “gut–brain axis modulation” and “role of bacteria in neuroinflammation”. All search results were manually reviewed prior to inclusion in this review. This review focuses mostly on original papers published within the past 5 years and/or studies that are highly relevant to the topic of the gut microbiota in Parkinsonian syndromes. Pilot studies and case reports were excluded from the analysis.

## 2. Role of the Intestinal Microbiota in Inflammation and PD Pathogenesis

The human digestive tract is a commonly known area of existence for about 100 trillion micro-organisms, including more than 160 bacterial species. The term “microbiota” refers to the group of living organisms existing in a certain environment, comprising bacteria, archaea, fungi, protists and algae, while not including phages, viruses, plasmids, prions, viroids, and free DNA [2]. Regarding bacteria, *Firmicutes*, *Bacteroidetes*, *Actinobacteria*, *Fusobacteria*, *Proteobacteria* and *Verrucomicrobia* make up the majority (i.e., up to 90%) of the intestinal microenvironment, with their concentration, composition and proportion differing among individuals, depending, i.e., on their localization in the gastrointestinal (GI) tract. The highest bacterial concentration is observed in the distal region (colon) [20]. At least 30% of the microbiota composition remains as a constant core group, while the remaining part is related to numerous factors such as lifestyle (stress, diet) and environment, as mentioned before [20,21].

The microbiota participates in crosstalk with the human body, forming an overarching system called a holobiont: the human digestive tract serves as a suitable environment for the sustainable development of bacteria and the production of numerous substances enabling the maintenance of homeostasis [22]. Interestingly, specific changes in the composition of microflora has been associated with the precise exacerbation of the clinical symptoms of PD—both cognitive and motor [23].

The availability of oxygen in the singular sections of the digestive tract strictly defines the type of bacteria living in that space. While *Lactobacillus* strains constitute the majority in the duodenum, anaerobic *Ruminococcaceae* and *Lachnospiraceae* predominate in the large intestine. Moreover, due to the presence of the intestinal barrier, bacteria that multiply and live in the intestinal lumen (LM) or in the mucous membrane (MAM) can be distinguished. Luminal bacteria in the duodenum are responsible for carbohydrate metabolism, while mucosal bacteria play a major role in endocrine system coordination. Large intestine luminal bacteria are focused mainly on lipid metabolism, while mucosal bacteria focus on general interactions with the host [24].

Inflammation—in addition to its undeniable impact in terms of increasing the permeability of the intestinal barrier—influences the selection of bacteria that are able to grow in the non-physiological conditions of this environment. Similarly to the blood–brain barrier, the integrity of the intestinal layer seems to act as a crucial factor protecting against pathogens and limiting the risk of toxic molecule absorption. The intestinal microbiota remains a key factor engaged with gut–brain axis activity. At present, numerous products of bacterial metabolic paths are being researched as factors that may reduce the risk of systemic inflammation and additional neurodegeneration [25].

Although the results of studies based on *Akkermansia* species remain contradictory, they are known as mucin-degrading bacteria, provoking a loss of intestinal barrier function while promoting gut inflammation and endotoxemia [26]. Nevertheless, in animal models, *Akkermansia muciniphila* has been shown to regulate the activity of T regulatory cells, short-chain fatty acid (SCFA) production and intestinal passage prolongation, as well as improving cognitive function by inhibiting the pro-inflammatory IL-6-dependent pathway [26,27]. Furthermore, the concentration of *Actinobacteria*—comprising, i.e., *Bifidobacterium* species—has been proven to be positively correlated with pro-inflammatory parameters, including white blood cells (WBCs) or the percentage of neutrophils and monocytes in the blood [28]. As these bacteria participate in histidine degradation and proline synthesis, they lead to increased glutamate levels in both pathways—thus promoting neuronal excitation—and increase the risk of excitotoxicity and neuronal cell death. Proline metabolism imbalance has been mentioned as a cause of alterations in neuronal activity and is linked with CNS diseases [29]. Arginine synthesis is strictly correlated with proline and glutamate acid concentrations. Its increased metabolism leads to the formation of putrescine and spermidine—molecules related to the degree of gut barrier permeability. Metagenomic studies have indicated that norspermidine enhances the aggregation of α-synuclein, while increased allantoin degradation could be a result of the inflammatory-based uric acid production, suggesting an increased risk of oxidative stress among PD patients [30]. Furthermore, some *Bifidobacteria* species contain succinate-semialdehyde dehydrogenase or glutamate decarboxylase, both resulting in an increased concentration of GABA, which is known as a promoter of PD motor symptoms [31].

The additional breakdown of connections between epithelial cells leads to increased exposure to lipopolysaccharides (LPS); that is, elements of the cellular wall of bacteria from the *Proteobacteria* group. This leads to a systemic pro-inflammatory reaction and the progression of neurodegeneration [32]. The excessive release of LPS and other bacterial metabolic products into the bloodstream could result in systemic inflammation progression, leading to disruption of the blood–brain barrier (BBB) and increasing the risk of enhanced transfer of inflammatory cells to the CNS, resulting in an increased risk of neuroinflammation and degeneration [16]. Recent data showed that LPS also increases the risk of vascular wall integrity disruption, leading to renal insufficiency and cardiovascular disease [33]. *Escherichia coli* and *Salmonella* are equipped with special fibers called Curli, which are the main proteinaceous component of their extracellular matrix (ECM) [34]. By binding with laminin and fibronectin, they provide contact with the host; however, they also bind t-PA, which activates plasminogen and leads to destruction of local tissue according to its protease property [35,36]. It is suspected that the Curli subunit CsgA (extracellular bacterial amyloid protein) interacts with mammalian proteins in so-called cross-seeding, promoting α-synuclein accumulation [37]. Interestingly, Curli belongs to the biofilm-associated proteins (BAPs) group, described as molecules that are responsible not only for the maintenance of biofilm integrity but also amyloid-like fibril accumulation, relentlessly leading to neurodegeneration [38].

Cells damaged by inflammation act as a great source of the host-derived immune-activating damage-associated molecular patterns (DAMPs), which are present in the blood and stimulate epithelial repair processes [39]. DAMPs show affinity with several types of receptors, such as toll-like receptors (TLRs), RIG-1-like receptors (RLS) and the Nod-like receptor family, triggering mechanisms of innate immunity. By stimulating myeloid-differentiation factor-88 and the NLRP3 inflammasome, they contribute to the activation of caspases and the release of pro-inflammatory interleukins, such as TNF-α, IL-1, IL-6, IL-8, and IL-12 via myeloid-differentiation factor-88 stimulation and IL-1β and IL-18 in the NLRP3 inflammasome [40].

Blood samples from PD patients have revealed increased levels of, i.e., IL-1β, IL-6 and TNF-α, while dysbiosis was also suspected to cause higher IFN-γ concentrations [41]. Comparing APS and idiopathic Parkinson’s disease (iPD) patients, the former revealed higher serum concentrations of IL-2, IL-4, IL-6, TNF-α, and IFN-γ, while the level of IL-10 was the lowest in APS patients [42].

Animal model-based research has revealed that a stress environment (water avoidance stress) leading to the inhibition of the NOD-like receptor and pyrin domain containing (NLRP)-6 inflammasome resulted in not only inflammatory enhancement, but also fundamental dysbiosis in mice [43].

A few articles have underlined the health-promoting role of inhalation of hydrogen sulfide (H2S), a substance produced mainly by *Prevotellaceae* (part of the *Proteobacteria* family) [44]. It is supposed to cause such a positive effect through the indirect expression of heme oxygenase-1 and glutamate-cysteine ligase, as well as anti-inflammatory properties [44]. However, trials in PD patients have shown that H2S enhances α-synuclein accumulation and could be triggered by *Desulfovibrionaceae*, as well as other bacteria, exposing PD patients to increased concentrations of this substance [45]. *Prevotellaceae* has been reported to decrease toxic polyglutamine (polyQ) aggregation and promote intestinal motility [46]. Some such species can also produce magnetite (Fe3O4), which collectively leads to excessive cytochrome c release in mitochondria, promotes iron accumulation and increases the level of reactive oxygen species production [47]. It has been postulated that rhamnolipid (RL)—an active element of the cellular surface produced by *Pseudomonas aeruginosa* (included in the *Proteobacteria* family)—could interact with α-synuclein molecules and indirectly modulate the risk of neurodegeneration in PD models [48]. Pseudomonas species also express genes for esterase PA2949, sharing a similar structure with the human membrane-bound α/β-hydrolase domain 6 (ABHD6) protein, leading to 2-arachidonylglycerol (2-AG) hydrolysis and endocannabinoid signaling impairment. As endocannabinoid receptors are mostly expressed on the surface of basal ganglia, their inhibition can result in impaired interactions with other pathways (including GABAergic and dopaminergic paths), increasing the risk of occurrence of neurodegeneration symptoms [49,50].

By reducing access to oxygen, anaerobic bacteria obtain optimal conditions to live and develop. The presence of *Actinobacteria* and *Enterococcus* species (e.g., *Olsenella*, *E. faecalis*) is associated with an increased level of the tyrosine metabolite p-cresol [51]. In a mouse model, p-cresol sulphate caused not only a decrease in circulating BDNF but also increased corticosterone and repressor element-1 silencing transcription factor (REST), leading to the development of depression-like, anxiety-like and cognitive impairment symptoms, including memory and learning difficulties [52]. Tyrosine decarboxylase (TyrDC) leads to premature synthesis of dopamine in the intestine, which is unable to cross the BBB and, thus, cannot have a beneficial effect in terms of alleviating the clinical symptoms of PD. Decreased levels of tyrosine, increased levels of α-synuclein, impaired sodium transport and higher succinate levels in feces and serum have been described in a mouse model of PD, notably revealing an increased concentration of *Alistipes* species [53].

Accordingly, isopropanol can have toxic effects on neuronal cells, being an alcohol that penetrates the CNS more effectively than ethanol and acting as a nervous system depressant [54]. Inhibition of its production concomitantly with alleviation of pyruvate fermentation to propanoate I have been shown to be facilitated by *Lactobacillus* (from phylum *Firmicutes*) [28]. As vitamin B12 is responsible for, i.e., propanoate breakdown, its potential protective role against inflammation and cell death, concomitantly with synaptic hippocampal support, has been shown in a rat model of AD [55].

A limited concentration of propanoate, caused by the mitochondrial unfolded protein response (mitoUPR) due to α-synuclein accumulation, has been shown to impede lipid and protein metabolism and impair energy production, which is crucial for effective gut–brain communication [56].

*Ruminococcaceae* bacteria (included in phylum Firmicutes) also exhibit numerous activities related to metabolic alterations [57]. They are predominantly responsible for α-linolenic acid (ALA) production, stimulating the Wnt/β-catenin signaling pathway and resulting in enhanced proliferation of intestinal stem cells [58]. *Ruminococcaceae* CPB6 have been found to express the LDH gene and promote apoptosis of microglial cells [59,60]. Apart from lactate modifications, CPB6 is also engaged in n-caproic acid (CA) production, involving other molecules, including acyl carrier protein (ACP) and coenzyme A (CoA) [61].

Toxins produced by particular *Clostridium* species (part of *Firmicutes*)—namely, through the inhibition of potassium channels—have been shown to lead to mitochondrial hyperpolarization, cellular death and increased intestinal permeability [62]. In a murine model, *Clostridium orbiscindens* has been shown to produce desaminotyrosine (DAT), which is also responsible for mucosal barrier maintenance and inflammatory attenuation. It is derived from tyrosine and flavonoid metabolism, and exhibits antiviral activity due to enhancement of the type I interferon signal-dependent pathway [63]. Reported data have indicated no significant change in microbiota diversity after levodopa treatment, although a reduced number of group IV *Clostridioides* bacteria has been described in patients showing significant motor improvement after levodopa initiation [64]. Similarly, regarding *Bifidobacteria*, participation in numerous metabolic paths related to amino acids allows *Lactobacillus* species to act as an important immunomodulatory element. The presence of *Lactobacillus* (belonging to *Firmicutes*) has been shown to correspond to the intensity of pyruvate fermentation to propanoate I and is negatively associated with isopropanol synthesis [28]. The ability of *L. plantarum* CCFM405 to promote the production of branched-chain amino acids (BCAAs)—including valine, leucine and isoleucine—has been proven to modify the composition of the intestinal microbiota, reduce inflammation (by decreasing IL-1β, IL-6 and TNF-α concentrations) and attenuate neuronal death [65]. Furthermore, it has been shown that another Lactobacillus representative (*L. plantarum PS128*) regulates improper microRNA (miR155-5p) expression and promotes the activity of a cytokine signaling inhibitor (SOCS1), exhibiting anti-inflammatory properties. While the gut microflora seems to be associated with the expression of numerous PD-related genes regulated by miRNAs, it is suspected that its diversity could act as a potential factor influencing the course of this disease [66].

The number of activities in which intestinal bacteria are involved emphasizes the importance of maintaining their proper composition, avoidance of dysbiosis and the provision of essential nutrients. Bacteria are involved in the formation of microbe–microbe and host–microbe interactions, indirectly contributing to the exacerbation or inhibition of neurodegeneration processes [67]. Through their local action, bacteria contribute to maintenance of the continuity of the intestinal barrier. Moreover, both direct molecules included in the bacterial cell membrane, as well as products of their metabolism, participate in numerous processes that are responsible for sustaining the energy balance—for example, lipid and carbohydrate metabolism, as well as immunomodulation and protection of the body against pathogens. SCFAs are synthesized through the fermentation processes carried out by bacteria; among these, acetate is involved in lipogenesis and gluconeogenesis, while butyrate and propionate are associated with the maintenance of physiological processes and regulation of the immune system [68].

Bacterial activity that is potentially important in the context of parkinsonian syndrome neurodegeneration is summarized in Table 1.

## 3. Role of the Intestinal Microbiota in Metabolism

### 3.1. SCFAs

Acetic acid (C2), propionic acid (C3) and butyric acid (C4) are the most abundant SCFAs in the human GI tract, with their concentrations being higher in its proximal part [51]. They occur in a ratio of 60:20:20 (C2:C3:C4), with C3 and C4 requiring the additional presence of the Firmicutes and Bacteroides family members within the intestine [69]. SCFAs serve as a crucial component regulating the microbiota–gut–brain axis, due to their numerous properties [70]. Butyrate’s beta-oxidation in mitochondria of intestinal epithelial cells ensures the continuity of intestinal barrier maintenance, providing, i.e., protection against pathogens [71,72]. SCFAs regulate the expression of genes encoding proteins (zonulin, occludin, claudin) that are part of tight junctions between enterocytes and stimulate the production of glycoproteins of the mucous layer [71]. They act as crucial factors coordinating processes involved in energy production and oxygen exploitation, regulating the level of hypoxia-inducible factor (HIF) and decreasing the risks of ROS synthesis, oxidative stress and intestinal barrier disruption [71]. Through regulation of the activities of enzymes such as tyrosine hydroxylase and tryptophan 5-hydroxylase 1, they take part in intermediate stages of the synthesis of tyrosine-related neurotransmitters—in particular, dopamine, adrenaline and noradrenaline—and are responsible for the conversion of tryptophan to serotonin [73]. In the available literature, 5-HT is known to impact lipid metabolism, regulate mitochondrial biogenesis and modulate the diversity of the intestinal microbiota [62]. The involvement of tyrosine metabolites, such as p-cresol and 4-ethylphenyl sulfate, in the gut–brain pathway has been described in the literature [74,75]. Tryptophan is mostly converted along the kynurenine pathway, which limits the extent to which it is metabolized into 5-HT. Once kynurenine has formed, it can act as a predictor of neuroprotection or neurotoxicity, as kynurenine’s metabolites—quinolinic and kynurenic acid—are NMDA receptor ligands reflecting opposite effects [76]. Tryptophan can also be metabolized into substances that stimulate Pregnane X receptors (PXRs), which participate in glucose and lipid metabolism processes and are indirectly activated by macrophages during inflammation [77,78,79]. Another tryptophan-derived substance is melatonin, which is responsible for coordination of the circadian rhythm and NF-κB-mediated immunomodulation [80,81].

SCFAs inhibit the accumulation of lipids in adipose tissue and participate in the regulation of carbohydrate metabolism, improving glucose tolerance [82]. They also possess neuroactive properties, and research in animal models has indicated the involvement of these acids in the maturation of microglial cells and maintaining the integrity of the BBB [73]. Through the regulation of neurotrophic factor levels, SCFAs have an impact on neurogenesis [83]. The concentration of brain-derived neurotrophic factor (BDNF) is proportionally related to the level of lactates, which are produced mainly by Lactobacillus and Bifidobacterium bacteria [83,84]. As LA increases the excitability of glutamatergic (NMDA) receptors, it contributes to an increase in intracellular calcium concentration and promotes the synthesis of BDNF [84]. Quinolinic acid, which participates in immune reactions, seems to modulate the astrocytic glutamate–glutamine cycle, simultaneously stimulating NMDA receptors and inhibiting glutamine synthetase [69]. It has been reported that neuronal activity enhanced by kainic acid (KA) receptors can be attenuated by melatonin in animal models, which could potentially become a future therapeutic approach in human trials [85].

SCFAs promote the differentiation of naive T lymphocytes into anti-inflammatory Tr1 and macrophage polarization into the M2 phenotype, sharing similar properties [86]. Through free fatty acid receptors (FFARs), they facilitate the reduction in pro-inflammatory iNOS and TNF-alpha [72]. They regulate gene expression, having the ability to inhibit histone deacetylases (HDACs), and are responsible for cellular signal transduction when attached to G protein-coupled receptors. It is suspected that they act as ligands of the aryl-hydrocarbon receptor—in animal models, AhR has been shown to be expressed on the surface of intraepithelial lymphocytes and is required to localize them to the skin and gut [71,72,87].

In terms of pathologic protein accumulation, another molecule called nerve growth factor (NGF) has been found to inhibit the hyperphosphorylation of tau protein in both in vitro and in vivo investigations, whereas glial cell line-derived neurotrophic factor (GDNF) is suspected to provide a therapeutic impact in PD patients, based on animal trials [73,88,89]. Notably, both of these factors are modulated by SCFAs [73].

Previous studies have shown that both the composition of the intestinal microbiota and the distribution of specific SCFAs differ depending on the biological age of the subjects [90]. This is undoubtedly influenced by the type of food consumed; for example, the composition of the intestinal microflora of breastfed infants differs significantly from that of humans on an adult diet mainly consisting of solid products [90,91]. With the end of the milk diet, the concentration of bacteria in the Firmicutes and Bacteroidetes phyla gradually increases, contributing to increases in the concentrations of propionate and butyrate [92]. Species of the genera Roseburia and Blautia—included in the Lachnospiraceae family (part of Firmicutes)—have been reported to participate in the process of intestinal maturation and exhibit immunomodulatory properties [41,93,94,95]. Lachnospiraceae are also responsible for secondary bile acid production and enteric pathogen inhibition [96].

### 3.2. Glucose and Lipid Metabolism

Existing data suggest that bacteria could influence both glucose and lipid metabolism. Interestingly, it is suspected that processes related to the transformation of carbohydrates are not always only a consequence of the implementation of overweight or anti-obesity treatments. Indirect intestinal bacterial activities mediated by SCFAs and succinate production could exhibit numerous metabolic effects [97].

A decreased level of insulin receptor phosphorylation, similar to the insulin receptor substrate (IRS) and Akt, could result in impaired insulin signaling, while some bacteria exhibit enzymatic activity including, i.e., glycoside hydrolases and carbohydrate esterases [98]. As SCFAs have been previously described to be molecules responsible for proper glucose metabolism, they also play a significant role in the regulation of lipid concentrations in the blood. Propionic acid, similar to commonly known statins, is known to inhibit 3-hydroxy-3-methyl glutaryl-coenzyme A synthase activity. Furthermore, a decreased concentration of reabsorbed bile salts in the liver resulting from the enzymatic activity of Lactobacilli species could impact the increased demand for bile salts and lead to enhanced cholesterol transformation, concurrently providing decreased blood cholesterol levels [98]. Data from ALS and AD revealed that the microbiota is also involved in polyunsaturated fatty acids (PUFAs), exhibiting both pro- and anti-inflammatory activity [99].

### 3.3. Vitamins

The intestinal microbiota creates a network exhibiting cross-feeding properties [100]. Available data suggest that at least 40% of gut species produce one of eight B-vitamins and are able to exchange vitamins among each other by complementary matching [101]. B vitamins are crucial not only for the production of ATP but also one-carbon (1C) metabolism, supporting the synthesis of purines (elements of nucleic acids) and the formation of amino acids (e.g., glycine, serine and methionine) [102].

As vitamin B1 (thiamine) participates in processes connected with ATP production and modulation of the immune system, there is a strong suspicion of its potential role in immunometabolism [103]. Research using a mouse model has revealed that thiamine deficiency—mostly produced by Prevotella, Desulfovibrio and Bacteroides—results in a significant reduction in Peyer’s patches and the number of B cells [102].

According to the significant number of vitamin B6 (pyridoxin) transporters in the large intestine, it is one of the few vitamins that is largely absorbed only in the distal part of the digestive tract [104]. It is responsible for the coordination of mucus production in the intestine, and acts as an antioxidant and cofactor in the production of neurotransmitters [105].

Vitamin B12 (VB12) is produced by only 20% of the intestinal microbiota [102]. It modulates the activity of epithelial cells and microbial functions in the intestines, exhibiting anti-inflammatory properties. It activates the HIF-1alpha pathway and, by enhancing mitochondrial processes, decreases the concentration of free oxygen within the intestines, creating a suitable environment for anaerobic bacteria and limiting the intensity of epithelial oxygenation [106]. VB12 also controls cellular proliferation, and its deficiency results in reduced villus height and a disruption of intestinal barrier maintenance [107]. A lack of VB12 leads to increases in kynurenine and homocysteine, resulting in increased inflammatory parameters [107,108].

Although there is no direct relationship between specific bacterial flora and the concentration of produced vitamin K, some studies have indicated a correlation between specific enterotypes (Prevotella and Bacteroides) and an increased amount of menaquinone in the stool [109]. Vitamin K and its active metabolites have been shown to reduce the concentrations of cytokines such as IL-1, IL-6 and TNF-alpha in both in vitro and in vivo investigations, indicating their potential anti-inflammatory properties [110]. The activation of Gla proteins affected the NF-kB/Nrf2 pathway (nuclear factor kappa-light-chain-enhancer of activated B cells (NF-kB)/nuclear factor erythroid 2-related factor 2) and decreased the risk of vascular inflammation in DM2 patients [111,112]. Additionally, it enabled carboxylation of the Gas6 protein (growth arrest-specific gene product 6), inhibiting neurodegeneration-related processes and limiting the cellular accumulation of free radicals. It indirectly inhibited certain aging processes (inflammaging) and epigenetic dysregulation, thus protecting against genomic instability [113,114].

The intestinal microbiota can also convert dietary vitamin A precursors into retinoic acid (RA), exhibiting immunomodulatory properties and acting as a regulator of gene expression [100,115]. The γt-orphan receptor related to the RA-receptor (RAR)—ROR—is expressed concurrently with AhR on the surface of lymphoid tissue-inducer cells, which are responsible for lymphoid development [116].

Impacts of bacterial metabolites on neurotransmitter production, microglial activation and blood–brain barrier integrity is summarized in Figure 1.

## 4. Role of the Intestinal Microbiota in Atypical Parkinsonian Syndromes

As mentioned before, limited data on divergence in the microbiota of APS patients has been described so far. According to the undoubted similarity when it comes to the clinical profiles of PD and APSs, comparisons of the microbiota profiles between these populations and those of HCs have been conducted. While the majority of these results overlapped, subtle differences were also found. Following Barichella et al., in the context of PD, Lachnospiraceae (in MSA) and Lactobacillaceae (in PSP) concentrations remained similar to those in HC, while Prevotellaceae (in MSA) and Streptococcaceae (in PSP) were decreased [23].

Results obtained in animal models have underlined the undeniable connection between immune system activity and neurodegeneration. Enhanced microgliosis and astrogliosis concurrent with the influx of T cells have been confirmed to play a role in tau-related neurodegeneration [117]. N6-carboxymethyllysine (CML)—one of the numerous microbial metabolites—is suspected to promote the enhancement of oxidative stress in microglia, resulting in mitochondrial impairment and the progression of intestinal permeability [118].

In vitro studies have revealed that the Gram-negative bacterium *Erwinia carotovora* 15 (Ecc15) could lead to regional inflammation, resulting in the release of numerous cytokines, activation of Janus kinase (JAK) receptors and subsequent promotion of the signal transducer and activator of transcription (STAT) signaling pathway. This leads to increased claudin-2 expression, which is involved in cellular tight junction regulation and increasing intestinal permeability via pore promotion [119,120]. 

Inhibition of Programmed Death protein 1 (PD-1), encoded by *Pdcd1*, has been reported to support long-term potentiation (LTP) and memory in hippocampal neurons [121]. Recent data have suggested that the suppression of PDCD1 could limit tauopathy-related neurodegeneration, while recent in vivo trials have revealed that the microbiota plays a role in the PD-1/PD-L1 pathway and exhibits immunomodulatory properties mediated by anti-PD1 molecules [122,123].

The *Streptococcaceae* concentration in PSP was found to be reduced, while there was no significant change in *Lactobacillaceae* [23]. Although the broad evaluation of fecal microbiota transplantation (FMT) treatments in PD patients has demonstrated the undeniable impact of the intestinal flora in immunomodulation and inflammatory alleviation, there has only been one published clinical trial suggesting a similar effect in PSP cases.

Existing studies point to the lack of a clear answer to the question regarding the mechanism of action of bacteria contributing to neuroprotection and inhibition of the accumulation of abnormal proteins. Studies in animal models have reported promising results, emphasizing the low risk of including human patients in similar studies. The results obtained to date have indicated a variety of results depending on the form of administration of the bacteria, which can be both oral and enteral [124].

After oral FMT in eight patients, a stool investigation revealed increased *Lactobacillaceae*, *Limnochordaceae* and *Peptostreptococcaceae* concentrations one week after therapy. One month after FMT, the patients reported improvements in areas such as motor performance, sleep quality and constipation; however, these improvements did not persist in later observations, which most likely indicates the need for regular FMTs in order to maintain the expected effects [124,125].

The results of FMT in the form of an oral capsule administered weekly for 3 weeks divided patients into responders and non-responders, according to the results described in terms of clinical scores. As a novel observation, the responders revealed increased concentrations of *Eubacterium*, *Clostridia* and *Roseburia* genera, similar to the donor’s microbiota diversity. The authors suggested that the response to FMT and engraftment success could be linked with (so far) unknown factors and underlined the importance of potential future biomarkers for the assessment of therapeutic effectiveness [126]. Intraintestinal microbiota administration indicated that FMT could also result in a decreased levodopa equivalent daily dose (LEDD) [127].

Restoring the healthy microbiota in PSP patients contributed to the alleviation of both motor and non-motor symptoms. While increased concentrations of *Escherichia-Shigella*, *Lactobacillus* and *Klebsiella* (in comparison with healthy controls, HC) were associated with a higher score in the PSPRS evaluation, PSP-RS patients also revealed decreased levels of *Faecalibacterium*, *Blautia*, *Agathobacter* (*Lachnospiraceae*), *Romboutsia* (*Peptostreptococcus*) and *Roseburia* species in comparison with HCs [128]. Nonetheless, the correlations between tau alterations and environmental factors such as dysbiosis still remain limited and further investigations in this context are required [117]. The precise link between microbiota and tauopathies can be described as misunderstood at present.

Under normal conditions, α-synuclein (α-Syn) is a presynaptically based unfolded protein divided into three regions, of which its central part presents hydrophobic properties and remains strictly fibrillogenic [129]. It is now known that improper forms of molecules, especially their aggregates, have a strong influence on microglial activation and could impede microglial autophagy [99]. The cyclic GMP–AMP synthase (cGAS)–stimulator of interferon genes (STING) signaling pathway, Toll-like receptor 4 (TLR4)-dependent p38 mitogen-activated protein kinase (MAPK) and AKT-mTOR cascade, tyrosine kinase-binding protein (TYROBP), triggering receptor expressed on myeloid cells 2-apolipoprotein E (TREM2-APOE) pathway, spleen tyrosine kinase (SYK), classical complement system, purinergic system, sialic acid binding immunoglobin-like lectins (Siglecs; CD22 and CD33), TAM system, and mechanosensor Piezo1 are only some of the potential mechanisms responsible for the regulation of microglial activity [99,129]. An increased concentration of isoamylamine (IAA)—related, i.e., with an increased abundance of *Ruminococcaceae*—consequently leads to deconfiguration of S100A8’s protein hairpin structure, enabling enhanced p53 activity [99]. Based on research in an animal model, the nucleotide-binding oligomerization domain leucine-rich repeat and pyrin domain-containing protein 3 (NLRP3) inflammasome also could potentially act as a molecule modulating the inflammatory response [99]. Decreased inflammasome-related caspase-1 concentration, concomitantly with decreased α-Syn, were found in all investigated PD cases, while their levels were positively correlated with monocyte activities [130].

α-Syn remains one of multiple factors responsible for monocyte activation. While it has not been profoundly explored, it exhibits a pro-inflammatory effect and differentiated cellular responses, possibly due to unexplored mechanisms [131]. Its level seems to be linked with the concentration of bacterial endotoxins and, although LPS can activate microglia and the pro-inflammatory M1 response, it can also induce the cellular death of astrocytes [130,132]. Monocyte Human Leukocyte Antigen (HLA)-DR expression has been found to be negatively correlated with motor and cognitive outcomes in PD patients [130].

The role of α-synuclein accumulation in MSA has been deeply investigated. As its strains are believed to cause disease-specific neurodegeneration, researchers have suggested the existence of a potential oligodendrocyte–neuron–α-Syn interplay, contributing to the development of MSA [133]. It is suspected that oligodendroglia cytoplasmic inclusions (GCIs) exhibit a distinct level of toxicity and their varied placement could serve as evidence of multiple symptoms. Interestingly, the amplification profile of α-synuclein derived from CSF in PD and MSA patients revealed different secondary structures, while its strains in DLB reflected traits of the ‘fibrils’ strain—in opposition to the ‘ribbons’ strain described previously in α-synucleinopathies.

Interestingly, recent studies have revealed that the E46K mutation in the α-synuclein gene inhibits neuropropagation of its self-templating prion conformation [134]. Animal studies have also shown that tau protein forms connections with α-synuclein, intensifying the process of their accumulation in PD [135].

MSA patients were found to have higher concentrations of *Prevotellaceae* concurrently with unchanged *Lachnospiraceae* levels [23]. A study in American MSA patients revealed higher *Clostridiaceae* and *Rikenellaceae* concentrations, while *Lachnospiraceae* (genera *Ruminococcus*, *Roseburia* and *Coprococcus*) and *Ruminococcaceae* (genus *Faecalibacterium*) were reduced [136]. An MSA patient in an ethnic Chinese population of Malaysia presented an increased level of *Bacteroides* and less-abundant *Paraprevotella* [137]. 

The described results seem to be inconclusive, due to the limited available patient groups as a result of their high morbidity and the rarity of the considered diseases [138]. Potential connections between gut microbiota and neurodegeneration are summarized in Figure 2.

## 5. Role of the Intestinal Microbiota in Future Diagnostic Processes and Treatments

In the context of the innovative idea of personalized medicine, there is still a need to identify readily available biological materials and highly specific markers with the aim of providing effective diagnostic approaches and successful treatments. Fecal investigations assessing abnormal concentrations of metabolites linked with riboflavin and biotin synthesis or carbohydrate-active enzymes (CAZymes) activity, which are responsible for carbohydrate degradation, or detecting a set of 11 genera with significant indications in PD using qPCR or 16S rRNA sequencing, could serve as examples of the abovementioned personalized approaches [139,140,141]. Metabolic alterations involving butyrate production, lipoic or alpha-linolenic acids, and glycerophospholipid metabolism are also linked with precise modifications in concrete bacterial species concentrations, which is considered to act as a potential determinant of more personal approaches for the diagnosis and treatment of PD [141]. Measurement of αSyn aggregation in the terminal ileum in combination with peripheral inflammation markers has also been underlined as a potential novel strategy for the identification of PD in its prodromal phase [142].

Elevated levels of peripheral αSyn could result in mitochondrial disruption, acting as a trigger for microglial activation and neuronal loss. Furthermore, the role of bacterial infection leading to an autoimmune mitochondria-specific response has also been underlined as a potential factor contributing to the development of PD, and it is considered that the mitochondrial lipid cardiolipin could induce enhanced oligomerization of pathological αSyn and intensify neuronal cell death. As different subtypes of Th17 cells are involved in inflammatory processes, it is supposed that bacteria could induce the specific switching between homeostatic Th17 and an inflammatory Th17 cell phenotype. Recent data have underlined the potential role of Bifidobacteria localized in the gut-associated mucosa; furthermore, animal studies have confirmed that an increased level of Enterobacteriacea crossing the intestinal barrier induces chronic inflammation via inflammatory Th17 cell phenotype activation.

Prebiotic/probiotic/synbiotic supplementation and fecal microbiome transplantations have been the most commonly investigated methods of microbial modification in PD trials [124]. Although the effects of FMT in PD have been deeply investigated, there is only one published study concerning its use in APS. So far, FMT in PSP has revealed promising results [126], contributing not only to improved physical activity but also attenuating gastrointestinal and psychiatric symptoms [128].

Prebiotics are molecules that act as substrates for the metabolic processes in which bacteria are directly involved. The most frequently studied prebiotics include dietary fiber (e.g., resistant starch). Most data have suggested PD-related dysbiosis, which led contemporary researchers to assess the effects of bacterial supplementation (in the form of probiotics) on the condition of patients suffering from PD. Recent results have indicated that three months of supplementation with *Lactobacillus acidophilus*, *Bifidobacterium bifidum*, *L. reuteri*, and *L. fermentum* contributed not only to the reduction in pro-inflammatory cytokines in the blood, but also influenced the motor symptoms of PD, as assessed according to the UPDRS scale [143,144]. The results obtained to date indicate that this therapy not only alleviates mild non-motor PD symptoms, but also acts as an immunomodulator, contributing to increased synthesis of SCFAs and reducing the intensity of inflammation—determined, for example, according to the concentration of calprotectin in stool [145,146].

To date, several studies taking into account the effects of probiotics on the clinical condition of patients with PD have been conducted. The mixtures administered most often consisted of *Lactobacillus* and/or *Bifidobacterium* strains; although their concentrations have been described in the literature as being increased in PD, the mechanism that leads to their expansion is not yet fully known [124]. In most cases, probiotic therapy significantly contributed to improved gastrointestinal function, while an improvement in motor skills (determined via UPDRS) was observed in only two of five studies [124]. Unfortunately, not all studies checked the final composition of the microbiota after the end of probiotic therapy. However, it is known that there is an increase in the number of fiber-fermentative bacteria, such as *Ruminococcaceae* and *Lachnospira*, and decreases in *Lactobacillus fermentum* and *Klebsiella oxytoca* [147,148].

Postbiotics are enzymes and substances that are the product of metabolic processes carried out by bacteria, as well as fragments of their cell membranes (e.g., polysaccharides, peptidoglycans, lipoic acid, phosphonic acid, cell surface proteins, cell membrane proteins and extracellular polysaccharides). Interestingly, an animal model study indicated that bacteria-derived lipids could cross the BBB and provide enhanced neurogenesis and inflammatory modulation [149]; furthermore, applying postbiotics alone did not reveal the same therapeutic effect as that observed after postbiotic injection [150].

Although there are many postbiotic molecules, most of them possess similar functions, such as reducing the level of oxidative stress, promoting the expression of anti-inflammatory cytokines and promoting neurogenesis [151].

It is worth mentioning that the term “Live Bacteriotherapeutic Products” (LBPs) seems to play a significant role when it comes to descriptions of potential innovative methods for the future treatment of PD [124]. As the field of personalized medicine is still dynamically developing, this term appears to be more appropriate.

The correlation between clinical PD symptoms and concentrations of inflammatory markers under an altered intestinal microbiota due to rifaximin usage has also been investigated [152]. Antibiotic treatment is being investigated in a clinical trial including patients with Parkinson’s disease dementia (PDD), who are planned to be treated with ceftriaxone [153]. Research based on animal models has shown that bacteriophage-delivered CRISPR/Cas9 endonuclease could promote the depletion of specific bacterial species and modify the microbiota profile [154].

Supplementation of anti-inflammatory molecules such as biotin or riboflavin, similarly to the insertion of butyrate producers (*B. producta*, inhibiting RAS-related pathways strictly correlated with inflammatory parameters in an animal model), has also been considered as an innovative method for the treatment of PD [139,155]. Interestingly, a mechanism of αSyn accumulation inhibition after *Lactobacillus plantarum*, *Clostridium butyricum* and *Bacillus subtilis* supplementation in an animal model was also underlined, which has not yet been certainly defined [124]. With an increased dietary intake of fiber concurrent with probiotic supplementation (called “synbiotics”), significant clinical changes have been described, such as, i.e., increased gut mobility [124]. Furthermore, intake of maltodextrin—a starch prescribed for the placebo group in one of the PD studies—was related to decreased UPDRS-III and increased MMSE scores [121].

## 6. Conclusions and Future Directions

APSs are a group of rare diseases with an etiology that has not been fully understood, and for which the methods of treatment developed so far have been ineffective. At present, researchers are still discovering new connections between neurodegeneration and numerous factors, the modification of which could affect the degree of severity of brain tissue damage. Some of the key elements of this complex system are inflammation and the diversity of intestinal microbiota, whose properties and connection with the brain via the bidirectional axis have been studied in the context of many other diseases.

It seems that the intensification of pro-inflammatory reactions, resulting from (among others) intestinal dysbiosis in patients with APSs, may affect the effectiveness of their treatment and constitute a factor that is potentially capable of modifying the course of the disease.

It cannot be absolutely excluded that future microbiome analyses may influence the selection of a proper treatment, or—if a cause–effect relationship was confirmed—that the modification of microbiota species may influence the course of the disease. Although APS and PD share certain similarities, the hypothesis suggesting a link between the gut microbiota and Parkinsonian syndromes is based on the α-synuclein pathology present in the enteric nervous system; therefore, the potential contribution of dysbiosis to tau pathology remains ambiguous.

In the authors’ opinion, future studies should focus on potential differences in the gut microbiota among distinct phenotypes of APSs and the exploration of potential connections between dysbiosis, inflammatory patterns and the course of the disease. Potential relationships between the course of the disease and the gut microbiota could initiate a new area of therapeutic approaches focused on non-invasive long-term interventions concerning the diet, probiotics and FMT, among others. Detailed analysis of the potential impacts of dysbiosis on neurodegeneration could lead to individualized treatments focused on the modification of the disease’s course; such a therapy could even be tailored to the individual patient, in the case of the importance of specific bacterial strains having been confirmed.

Although promising, all of the above-described hypotheses still require in-depth research for confirmation.

## Figures and Tables

**Figure 1 ijms-26-03928-f001:**
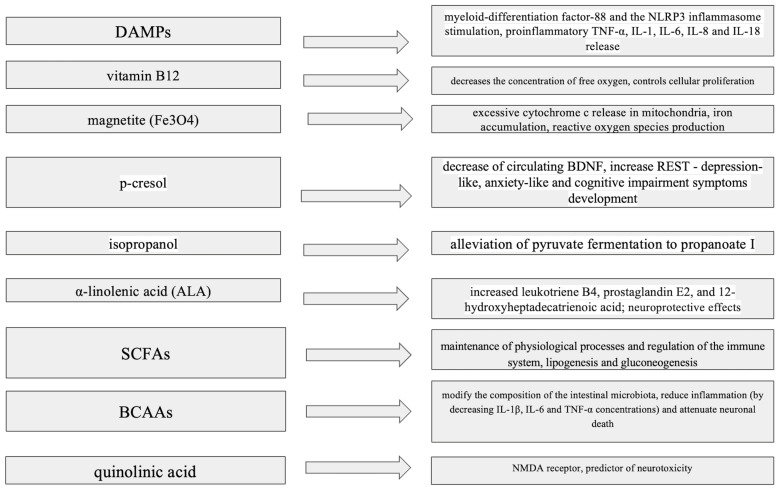
Impact of bacterial metabolites on neurotransmitter production, microglial activation and blood–brain barrier integrity. ALA—α-Linolenic Acid, BCAAs—Branched-Chain Amino Acids, BDNF—Brain-Derived Neurotrophic Factor, DAMPs—Damage-Associated Molecular Patterns, IL-1—Interleukin-1, IL-6—Interleukin-6, IL-8—Interleukin-8, IL-18—Interleukin-18, NLRP3—NOD-like Receptor Protein 3, NMDA—N-Methyl-D-Aspartate, p-cresol—Para-Cresol, REST—Repressor Element-1 Silencing Transcription Factor, SCFAs—Short-Chain Fatty Acids, TNF-α—Tumor Necrosis Factor Alpha.

**Figure 2 ijms-26-03928-f002:**
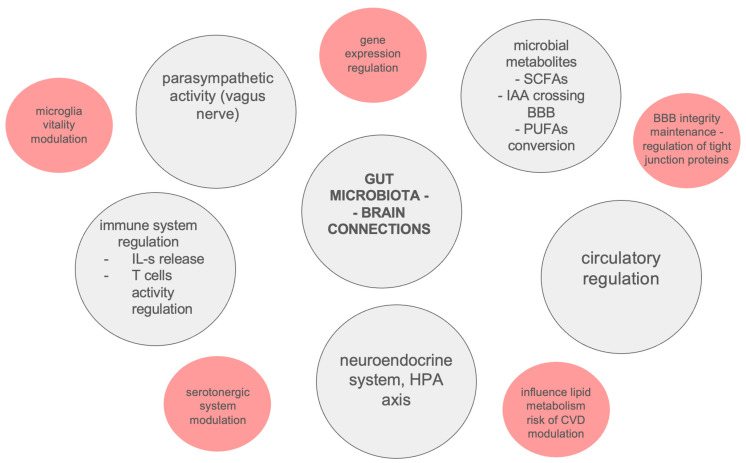
Potential connections between gut microbiota and neurodegeneration. BBB—Blood–Brain Barrier, CVD—Cardiovascular Disease, HPA—Hypothalamic–Pituitary–Adrenal axis, IAA—Isoamylamine, IL-s—Interleukins, PUFAs—Polyunsaturated Fatty Acids, SCFAs—Short-Chain Fatty Acids.

**Table 1 ijms-26-03928-t001:** Bacterial activity potentially influencing neurodegeneration.

Bacteria	Activity	Effect
*Akkermansia*	degrades mucin regulates T regulatory cell activity regulates SCFA production inhibits IL-6 dependent pathway	intestinal passage prolongation cognitive function improvement
*Actinobacteria*	increases glutamate level, promoting neuronal excitation forms putrescine and spermidine	enhancement of aggregation of α-synuclein risk of increased oxidative stress
*Desulfovibrionaceae*	produces magnetite (Fe_3_O_4_)	excessive cytochrome c release in mitochondria iron accumulation and increase in the level of reactive oxygen species production
*Proteobacteria*	rhamnolipid (RL) interacts with α-synuclein molecules disables endocannabinoid receptors	risk of neurodegeneration symptoms occurrence
*Prevotellaceae*	decreases toxic polyglutamine (polyQ) aggregation	decreased constipation
*Firmicutes* (*type*) *Bacilli* (*class*) *Lactobacillus* *Enterococcus* *Streptococcaceae* *Clostridium* (*class*) *Lachnospiraceae* *Ruminococcaceae*	inhibits isopropanol production ferments pyruvate to propanoate I promotes BCAAs production produces p-cresol decreases circulating BDNF disrupts intercellular connections inhibits potassium channels produces SCFAs produces secondary bile acids produces α-linolenic acid (ALA)	potential protective role against inflammation and cell death (in AD) intestinal microbiota composition modification inflammation reduction depression-like, anxiety-like and cognitive impairment symptoms development increased intestinal permeability increased cellular death and intestinal permeability enteric pathogen inhibition mucosal immunity regulation apoptosis of microglial cells

Legend: AD—Alzheimer’s Disease, ALA—α-Linolenic Acid, BCAAs—Branched-Chain Amino Acids, BDNF—Brain-Derived Neurotrophic Factor, IL-6—Interleukin-6, p-cresol—Para-Cresol, polyQ—Polyglutamine, RL—Rhamnolipid, SCFAs—Short-Chain Fatty Acids.

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
