# Peer review of "Role of the Intestinal Microbiota in the Molecular Pathogenesis of Atypical Parkinsonian Syndromes"

_ijms, 2025, doi:10.3390/ijms26093928_

Round 1

Reviewer 1 Report

Comments and Suggestions for Authors

The manuscript by Dominika Przewodowska et al. summarized recent advances in the study of the role of gut microbiota in the pathogenesis of parkinsonian syndromes. The review is basically good. I have the following questions and comments.

1, the title of the manuscript could be changes to "xxx pathogenesis of atypical parkinsonian syndromes". 

2, the authors must specify how the literature was obtained and any keywords used while searching the database. The inclusion and exclusion criteria must be provided. 

3, for the discussion of the SCFAs, I suggest the authors to introduce more about the bacteria that could produce SCFAs. Besides, any bacteria that are beneficial for the prevention of PD should also be discussed. 

4, a table summarizing recent advances in the clinical study should be provided. 

5, the authors should also discuss more about the potential effects of FMT, prebiotics, probiotics, and postbiotics on the development of PD. 

6, a new figure illustrating the molecular mechanisms underlying the the role of gut microbiota in the pathogenesis of parkinsonian syndromes must be provided. 

7, the global epidemiology and disease burden of PD should also be discussed and introduced. 

Author Response

Dear Reviewer 1, 

We are very grateful for your valuable comments. All suggestions were implemented, we believe that current version of manuscript improved significantly and is suitable for IJMS. 

1, the title of the manuscript could be changes to "xxx pathogenesis of atypical parkinsonian syndromes". 

The title was changed according to the suggestion.

2, the authors must specify how the literature was obtained and any keywords used while searching the database. The inclusion and exclusion criteria must be provided. 

Methods of literature search and inclusion criteria were specified – a paragraph:
“This paper provides a narrative review concerning potential role of gut microbiota in pathogenesis of atypical parkinsonian syndromed. PubMed and Google Scholar databases were searched for papers related to the molecular mechanisms of bacterial activity in atypical parkinsonism. The following medical subject headings were used: "atypical parkinsonian syndromes-role of intestinal microbiota," "molecular mechanisms of neurodegeneration," "microbiota activity," "gut-brain axis activity”, “gut-brain axis modulation”, “role of bacteria in neuroinflammation”. All search results were manually reviewed prior to inclusion in this review. The review focuses mostly on original papers published during last 5 years or studies most relevant to the topic of gut microbiota in parkinsonian syndromes. Pilot studies or case-reports were excluded from the analysis.” was added.

3, for the discussion of the SCFAs, I suggest the authors to introduce more about the bacteria that could produce SCFAs. Besides, any bacteria that are beneficial for the prevention of PD should also be discussed. 

Paragraphs:

"The number of activities in which intestinal bacteria are involved emphasizes the importance of maintaining their proper composition, avoidance of dysbiosis and the provision of essential nutrients. Bacteria are involved in the formation of microbemicrobe and hostmicrobe interactions, indirectly contributing to the exacerbation or inhibition of neurodegeneration processes [67].Through their local action, bacteria contribute to maintenance of the continuity of the intestinal barrier. Moreover, both direct molecules included in the bacterial cell membrane, as well as products of their metabolism participate in numerous processes which are responsible for sustaining the energy balance—for example, lipid and carbohydrate metabolism, as well as immunomodulation and protection of the body against pathogens. SCFAs are synthesized through the fermentation processes carried out by bacteria; of these, acetate is involved in lipogenesis and gluconeogenesis, while butyrate and propionate are associated with the maintenance of physiological processes and regulation of the immune system [68]."

and:

"Previous studies have shown that both the composition of the intestinal microbiota and the distribution of specific SCFAs differ depending on the biological age of the subjects [90]. This is undoubtedly influenced by the type of food consumed; for example, the composition of the intestinal microflora of breastfed children differs significantly from that of patients on an adult diet mainly consisting of solid products [90, 91]. With the end of the milk diet, the concentration of bacteria in the Firmicutes and Bacteroidetes phyla gradually increases, contributing to increases in the concentrations of propionate and butyrate [92]. Species of the genera Roseburia and Blautia—included in the Lachnospiraceae family (part of Firmicutes)—have been described to participate in the process of intestinal maturation and exhibit immunomodulatory properties [41, 93-95]. Lachnospiraceae are also responsible for secondary bile acids production and enteric pathogens inhibition [96].",

and:

"Most data have indicated PD-related dysbiosis, which led contemporary researchers to assess the effects of bacterial supplementation (in the form of probiotics) on the condition of patients suffering from PD. Recent results have indicated that three months of supplementation with Lactobacillus acidophilus, Bifidobacterium bifidum, L. reuteri, and L. fermentum contributed not only to the reduction in pro-inflammatory cytokines in the blood, but also influenced the motor symptoms of PD, as assessed according to the UPDRS scale [143, 144]. The results obtained to date indicate that this therapy not only alleviates mild non-motor PD symptoms, but also acts as an immunomodulator, contributing to increased synthesis of SCFAs and reducing the intensity of inflammation—determined, for example, according to the concentration of calprotectin in stool [145, 146].

To date, several studies taking into account the effects of probiotics on the clinical condition of patients with PD have been conducted. The mixtures administered most often consisted of Lactobacillus and/or Bifidobacterium strains; although their concentrations have been described in the literature as being increased in PD, the mechanism that leads to their expansion is not yet fully known [124]. In most cases, probiotic therapy significantly contributed to improved gastrointestinal function, while an improvement in motor skills (determined via UPDRS) was observed in only 2 of 5 studies [124]. Unfortunately, not all studies checked the final composition of the microbiota after the end of probiotic therapy. However, it is known that there is an increase in the number of fiber-fermentative bacteria, such as Ruminococcaceae and Lachnospira, and decreases in Lactobacillus fermentum and Klebsiella oxytoca [147, 148]."

were added.

4, a table summarizing recent advances in the clinical study should be provided. 

A table (table 1) summarizing bacterial activity potentially important in the context of parkinsonian syndromes neurodegeneration was added. 

5, the authors should also discuss more about the potential effects of FMT, prebiotics, probiotics, and postbiotics on the development of PD. 

Paragraphs:

"Existing studies point to the lack of a clear answer to the question regarding the mechanism of action of bacteria contributing to neuroprotection and inhibition of the accumulation of abnormal proteins. Studies in animal models have reported promising results, emphasizing the low risk of including human patients in similar studies. The results obtained to date have indicated a variety of results depending on the form of administration of the bacteria, which can be both oral and enteral [124].

After oral FMT in eight patients, a stool investigation revealed increased Lactobacillaceae, Limnochordaceae and Peptostreptococcaceae concentrations one week after therapy. At a month after therapy, they reported improvements in areas such as motor performance, sleep quality and constipation; however, these improvements did not persist in later observations, which most likely indicates the need for regular visits and FMT in order to maintain the expected effects [124, 125].

The results of FMT in the form of an oral capsule administered weekly for 3 weeks divided patients into responders and non-responders, according to the results described in terms of clinical scores. As a novel observation, the responders revealed increased concentrations of Eubacterium, Clostridia and Roseburia genera, similarly to the donor's microbiotal diversity. The authors suggested that the response to FMT and engraftment success could be linked with (so far) unknown factors and underlined the importance of potential future biomarkers for the assessment of therapeutic effectiveness [126]. Intraintestinal microbiota administration indicated that FMT could also result in a decreased levodopa equivalent daily dose (LEDD) [127]."

 and:

"Prebiotics are molecules that act as substrates for the metabolic processes in which bacteria are directly involved. The most frequently studied prebiotics include dietary fiber (e.g., resistant starch). Most data have indicated PD-related dysbiosis, which led contemporary researchers to assess the effects of bacterial supplementation (in the form of probiotics) on the condition of patients suffering from PD. Recent results have indicated that three months of supplementation with Lactobacillus acidophilus, Bifidobacterium bifidum, L. reuteri, and L. fermentum contributed not only to the reduction in pro-inflammatory cytokines in the blood, but also influenced the motor symptoms of PD, as assessed according to the UPDRS scale [143, 144]. The results obtained to date indicate that this therapy not only alleviates mild non-motor PD symptoms, but also acts as an immunomodulator, contributing to increased synthesis of SCFAs and reducing the intensity of inflammation—determined, for example, according to the concentration of calprotectin in stool [145, 146].

To date, several studies taking into account the effects of probiotics on the clinical condition of patients with PD have been conducted. The mixtures administered most often consisted of Lactobacillus and/or Bifidobacterium strains; although their concentrations have been described in the literature as being increased in PD, the mechanism that leads to their expansion is not yet fully known [124]. In most cases, probiotic therapy significantly contributed to improved gastrointestinal function, while an improvement in motor skills (determined via UPDRS) was observed in only 2 of 5 studies [124]. Unfortunately, not all studies checked the final composition of the microbiota after the end of probiotic therapy. However, it is known that there is an increase in the number of fiber-fermentative bacteria, such as Ruminococcaceae and Lachnospira, and decreases in Lactobacillus fermentum and Klebsiella oxytoca [147, 148].

Postbiotics are enzymes and substances that are the product of metabolic processes carried out by bacteria, as well as fragments of their cell membranes (e.g., polysaccharides, peptidoglycans, lipoic acid, phosphonic acid, cell surface proteins, cell membrane proteins and extracellular polysaccharides). Interestingly, an animal-based trial indicated that bacteria-derived lipids could cross the BBB and provide enhanced neurogenesis and inflammatory modulation [149]; furthermore, applying postbiotics alone did not reveal the same therapeutic effect as that observed after postbiotic injection [150].

Although there are many postbiotic molecules, most of them possess similar functions, such as reducing the level of oxidative stress, promoting the expression of anti-inflammatory cytokines and promoting neurogenesis [151]." were added. 

6, a new figure illustrating the molecular mechanisms underlying the the role of gut microbiota in the pathogenesis of parkinsonian syndromes must be provided. 

A figure summarizing impact of bacterial metabolites on neurotransmitter production, microglial activation and blood-brain barrier integrity (figure 1) and figure summarizing potential connections between gut microbiota and neurodegeneration (figure 2) were added. 

7, the global epidemiology and disease burden of PD should also be discussed and introduced. 

Suggestion impelemented - "While the prevalence of Parkinsons disease is expected to increase from 76 (in 2021) to 267 cases per 100,000 in 2050, the prevalence of APSs tends to remain stable over time and has been estimated as less than 10 per 10,000 [8, 9]" was added. 

With best regards

Natalia Madetko-Alster

Reviewer 2 Report

Comments and Suggestions for Authors

The manuscript entitled „Potential role of gut microbiota in the pathogenesis of parkinsonian syndromes” by Przewodowska et al. is well-structured and provides a comprehensive review of the role of gut microbiota in the pathogenesis of atypical parkinsonian syndromes. However, there are a few areas that could be improved:

There are some grammatical inconsistencies and awkward phrasings throughout the manuscript.

Example: "Although Braak’s (dual-hit) hypothesis of sporadic PD development is commonly known since almost 20 years, it’s accuracy is still debated."

Suggestion: "Although Braak’s dual-hit hypothesis of sporadic PD development has been widely recognized for nearly 20 years, its accuracy remains debated."

The transition between sections could be improved for a smoother flow. For example, the discussion of inflammation mechanisms could be better linked to the subsequent sections on microbiota and neurodegeneration.

The conclusion is informative but could be more forward-looking. Consider adding a section on potential future research directions or clinical applications.

The manuscript covers a wide range of topics related to gut microbiota and atypical parkinsonian syndromes, making it well-suited for visual representation. Here are some suggested figures that could enhance clarity and engagement: A bar graph or pie chart comparing bacterial genera found in: healthy individuals, Patients with PD, Patients with atypical Parkinsonian syndromes (MSA, PSP, CBS); A diagram showing how bacterial metabolites (SCFAs, p-cresol, LPS, quinolinic acid, etc.) affect neurotransmitter production, microglial activation, blood-brain barrier integrity; schematic showing the pathways connecting the gut microbiota to the brain….

Comments on the Quality of English Language

The language might be improved. 

There are some grammatical inconsistencies and awkward phrasings throughout the manuscript.

Example: "Although Braak’s (dual-hit) hypothesis of sporadic PD development is commonly known since almost 20 years, it’s accuracy is still debated."

Suggestion: "Although Braak’s dual-hit hypothesis of sporadic PD development has been widely recognized for nearly 20 years, its accuracy remains debated."

Author Response

Dear Reviewer 2, 

We are very grateful for your valuable comments. All suggestions were implemented, we believe that current version of manuscript improved significantly and is suitable for IJMS. 

There are some grammatical inconsistencies and awkward phrasings throughout the manuscript.
The manuscript was reviewed by English native speaker - certificate is attached in additional files. 

The transition between sections could be improved for a smoother flow. For example, the discussion of inflammation mechanisms could be better linked to the subsequent sections on microbiota and neurodegeneration.

The manuscript was reviewed by English native speaker, transition between sections was improved.

The conclusion is informative but could be more forward-looking. Consider adding a section on potential future research directions or clinical applications.

Paragraph concerning future research directions and clinical applications was added.

The manuscript covers a wide range of topics related to gut microbiota and atypical parkinsonian syndromes, making it well-suited for visual representation. Here are some suggested figures that could enhance clarity and engagement: A bar graph or pie chart comparing bacterial genera found in: healthy individuals, Patients with PD, Patients with atypical Parkinsonian syndromes (MSA, PSP, CBS); A diagram showing how bacterial metabolites (SCFAs, p-cresol, LPS, quinolinic acid, etc.) affect neurotransmitter production, microglial activation, blood-brain barrier integrity; schematic showing the pathways connecting the gut microbiota to the brain….

Two figures (1. Impact of bacterial metabolites on neurotransmitter production, microglial activation and blood-brain barrier integrity, 2.Potential connections between gut microbiota and neurodegeneration) and one table (Bacterial activity potentially influencing neurodegeneration) were added. 

With best regards
Natalia Madetko-Alster

Reviewer 3 Report

Comments and Suggestions for Authors

      This is a review paper that analyzes the possible role of intestinal microbiota in the pathogenesis of atypical parkinsonian syndromes. In particular, the authors present the known pathophysiological mechanisms and theories in parkinsonian syndromes. Moreover, they examine the role of inflammation in initiating neurodegeneration and its relation with intestinal microbiota.  In addition, the paper focus on the description of possible associations of microbiota composition and/or metabolic functions on neurodegeneration and in particular on PD, which comprises the biggest part of the text, and atypical parkinsonian syndromes evolution and progression. Vitamins’ contribution is an interesting part of the review.

In general, this paper content is of high quality and meets the publication standards of a narrative review, but it suffers from some flaws, especially concerning the structure of the manuscript and the presentation of some important aspects of the topic.

 Comments

 Major concerns:

    My main concern is that the structure of the manuscript should be modified. Although, this paper content is of high scientific value and references are recent and in most cases adequate, the presentation of the topic is rather confusing for the reader. The material is presented in an ‘‘ungrouped” manner, which is not easy to follow.

The addition of some figures might be helpful.

I suggest to reorganize the introduction. The authors should add here a paragraph describing the physiological role and composition of gut microbiota instead of the main text and refer briefly some key metabolic functions. Moreover, some basic definitions should be added here (eg Microbiota, dysbiosis etc). Revision of the main text is recommended as well. 

Minor concerns:

Introduction

  1. Page 1, lines 28-32: “Atypical parkinsonian syndromes (APS) comprise of various neurodegenerative disorders characterized by rapid progression, poor levodopa responsiveness and shorter life expectancy compared to Parkinson’s disease (PD). Among them can be distinguished Progressive Supranuclear Palsy (PSP), Multiple System Atrophy (MSA) and Corticobasal Syndrome (CBS)”. The authors should add appropriate references, which are missing.
  2. The authors should briefly describe the basic clinical features of the atypical parkinsonism syndromes and refer to the similarities and differences. Also, some authors include to parkinsonian syndromes the Lewy body disease (DLB). They should explain why they have excluded this clinical entity.
  1. In the context of the previous comment and concerning the pathophysiology of atypical parkinsonian syndromes, I suggest to expand this part of the introduction. In particular, there is no mention that the parkinsonian syndromes are characterized as proteinopathies, because they are associated with extra- and intracellular accumulation of misfolded proteins. The authors mention the pathophysiology of PD (including Braak theory) but they do not adequately explain the pathophysiology of atypical parkinsonian syndromes. Thus, it is important to highlight that their pathological hallmarks are different: PD and MSA are synucleinopathies, due to aggregation of α-synuclein protein and the PSP and CBD are tauopathies. This indicates that may not share the same pathophysiology. However, in the main text there are scattered reports to these differences, which should be reorgorganised.
  1. Page 2, lines 66-67: References are missing. I suggest to add proper literature justifying the content of the text.

Main text

  1. Another important issue is that, although the title refers to atypical parkinsonian syndromes, the majority of the main text concerns in fact Parkinson’s disease as literature in atypical parkinsonism and gut microbiota is limited.
  1. The authors should describe in the introduction that short chain fatty acids (SCFAs) are microbial metabolites.
  2. Page 8, lines 354-356: "Following Barichella et al. results - differently as in PD - Lachnospiraceae (in MSA) and Lactobacillaceae (in PSP) concentration remained similar as in HC, while Prevotellaceae (in MSA) and Streptococcaceae (in PSP) were decreased [92]". The reference is not the proper one, which is reference number 14.
  3. Page 8, lines 374-375: "Streptococcaceae concentration in PSP was found to be reduced with no significant 374 change in Lactobacillaceae [92]". Wrong Referernce again
  4. Page 8-9, lines 387-413: These paragraphs contain general statements about a synuclein or about its role in PD, nothing in specific about MSA, Should be referred elsewhere.
  5. Page 9, lines 415-416: "MSA patients were found to have higher concentration of Prevotellaceae concurrently with unchanged Lachnospiraceae level [92]". Referernce
  6. Page 9, lines 422: Comparison of specific microbiota components for PSP and MSA is presented in table 1. The authors should clarify if there any data available for CBD as well. In general, besides the introductory paragraph, CBD is not described elsewhere in the text.

Conclusions

  1. I suggest to clearly state that atypical parkinsonian syndromes are characterized by similarities and differences and that the hypothesis of the link between microbiota and parkinsonism has been based on the aggregation of α-synuclein in the enteric nervous system and nothing is known yet about contribution of tau pathology.

Author Response

Dear Reviewer 3, 

We are very grateful for your valuable comments. All suggestions were implemented, we believe that current version of manuscript improved significantly and is suitable for IJMS. 

 Major concerns:

    My main concern is that the structure of the manuscript should be modified. Although, this paper content is of high scientific value and references are recent and in most cases adequate, the presentation of the topic is rather confusing for the reader. The material is presented in an ‘‘ungrouped” manner, which is not easy to follow.

The manuscript structure was reorganized. 

The addition of some figures might be helpful.

Two figures and one table were added.

I suggest to reorganize the introduction. The authors should add here a paragraph describing the physiological role and composition of gut microbiota instead of the main text and refer briefly some key metabolic functions. Moreover, some basic definitions should be added here (eg Microbiota, dysbiosis etc). Revision of the main text is recommended as well. 

The introduction was reorganized and main text was revised according to the suggestion. 

Minor concerns:

Introduction

  1. Page 1, lines 28-32: “Atypical parkinsonian syndromes (APS) comprise of various neurodegenerative disorders characterized by rapid progression, poor levodopa responsiveness and shorter life expectancy compared to Parkinson’s disease (PD). Among them can be distinguished Progressive Supranuclear Palsy (PSP), Multiple System Atrophy (MSA) and Corticobasal Syndrome (CBS)”. The authors should add appropriate references, which are missing.
    References implemented. 
  2. The authors should briefly describe the basic clinical features of the atypical parkinsonism syndromes and refer to the similarities and differences. Also, some authors include to parkinsonian syndromes the Lewy body disease (DLB). They should explain why they have excluded this clinical entity.
      APSs were briefly described. 
    DLB is usually discussed in the context of dementia, moreover it is less challenging in terms of differential diagnosis. What is more, it is still debated, if DLB and Parkinson’s disease dementia (PDD) should be described as separate entities at all. Taking into consideration above mentioned uncertainties, authors decided to exclude DLB from the review.
  1. In the context of the previous comment and concerning the pathophysiology of atypical parkinsonian syndromes, I suggest to expand this part of the introduction. In particular, there is no mention that the parkinsonian syndromes are characterized as proteinopathies, because they are associated with extra- and intracellular accumulation of misfolded proteins. The authors mention the pathophysiology of PD (including Braak theory) but they do not adequately explain the pathophysiology of atypical parkinsonian syndromes. Thus, it is important to highlight that their pathological hallmarks are different: PD and MSA are synucleinopathies, due to aggregation of α-synuclein protein and the PSP and CBD are tauopathies. This indicates that may not share the same pathophysiology. However, in the main text there are scattered reports to these differences, which should be reorgorganised.
    Suggestion implemented, introduction was extended, main text reorganized according to the suggestion. 
  1. Page 2, lines 66-67: References are missing. I suggest to add proper literature justifying the content of the text.
    References were added.

Main text

  1. Another important issue is that, although the title refers to atypical parkinsonian syndromes, the majority of the main text concerns in fact Parkinson’s disease as literature in atypical parkinsonism and gut microbiota is limited.
    Unfortunately, current studies concerning gut microbiota are focused on PD patients and relatively small number of studies included patients with APS. Authors of the review intended to highlight potential impact of gut microbiota in APS by analyzing mechanisms potentially impacting neurodegeneration by analogy to what is already known in the context of PD.
  1. The authors should describe in the introduction that short chain fatty acids (SCFAs) are microbial metabolites.
    Suggestion implemented.
  2. Page 8, lines 354-356: "Following Barichella et al. results - differently as in PD - Lachnospiraceae (in MSA) and Lactobacillaceae (in PSP) concentration remained similar as in HC, while Prevotellaceae (in MSA) and Streptococcaceae (in PSP) were decreased [92]". The reference is not the proper one, which is reference number 14.
    Change implemented. 
  3. Page 8, lines 374-375: "Streptococcaceae concentration in PSP was found to be reduced with no significant 374 change in Lactobacillaceae [92]". Wrong Referernce again
    Change implemented. 
  4. Page 8-9, lines 387-413: These paragraphs contain general statements about a synuclein or about its role in PD, nothing in specific about MSA, Should be referred elsewhere.
    Change implemented. 
  5. Page 9, lines 415-416: "MSA patients were found to have higher concentration of Prevotellaceae concurrently with unchanged Lachnospiraceae level [92]". Referernce
    Change implemented. 
  6. Page 9, lines 422: Comparison of specific microbiota components for PSP and MSA is presented in table 1. The authors should clarify if there any data available for CBD as well. In general, besides the introductory paragraph, CBD is not described elsewhere in the text.
    A paragraph :"Neurocognitive dysfunction—often misdiagnosed as AD—is a common sign of CBS. Coincidentally, it is also characterized by alien limb phenomena, described as asymmetric rigidity, hypokinesia and dystonia with numerous uncontrolled limb movements [2]. At present, there are no data concerning dysbiosis in corticobasal degeneration (CBD), which could be (at least partially) explained by the rarity of the disease and the need for neuropathological examination in order to obtain diagnosis. CBS, which can be diagnosed based on the clinical syndrome, is very diverse in the context of neuropathology, which could explain the lack of studies concerning the gut microbiota with respect to this disease." was added.

Conclusions

  1. I suggest to clearly state that atypical parkinsonian syndromes are characterized by similarities and differences and that the hypothesis of the link between microbiota and parkinsonism has been based on the aggregation of α-synuclein in the enteric nervous system and nothing is known yet about contribution of tau pathology.Change implemented, conclusions were modified. 

With best regards
Natalia Madetko-Alster

Round 2

Reviewer 1 Report

Comments and Suggestions for Authors

The authors have revised the manuscript accordingly. It can be considered for publication. 

Reviewer 2 Report

Comments and Suggestions for Authors

The authors addressed all my concerns. 

Reviewer 3 Report

Comments and Suggestions for Authors

This revised manuscript has been significantly improved. The authors have satisfactorily addressed my comments. The introduction and main text of the paper have been efficiently reorganized and all required changes have been included in the revised manuscript. Added table and figures are interesting and reader friendly. I only suggest authors to include as well the table named as "Microbiota components – PSP and MSA comparison" in the final form.